# Monitoring Vitamin B_12_ in Women Treated with Metformin for Primary Prevention of Breast Cancer and Age-Related Chronic Diseases

**DOI:** 10.3390/nu11051020

**Published:** 2019-05-07

**Authors:** Antonio Mastroianni, Chiara Maura Ciniselli, Rossella Panella, Alessandra Macciotta, Adalberto Cavalleri, Elisabetta Venturelli, Francesca Taverna, Arabella Mazzocchi, Eleonora Bruno, Paola Muti, Franco Berrino, Paolo Verderio, Daniele Morelli, Patrizia Pasanisi

**Affiliations:** 1Laboratory Medicine-Department of Pathology and Laboratory Medicine, Fondazione IRCSS Istituto Nazionale dei Tumori di Milano, 20133 Milano, Italy; rossella.panella@istitutotumori.mi.it (R.P.); daniele.morelli@istitutotumori.mi.it (D.M.); 2Bioinformatics and Biostatistics Unit-Department of Applied Research and Technological Development, Fondazione IRCSS Istituto Nazionale dei Tumori di Milano, 20133 Milano, Italy; chiara.ciniselli@istitutotumori.mi.it (C.M.C.); alessandra.macciotta@istitutotumori.mi.it (A.M.); paolo.verderio@istitutotumori.mi.it (P.V.); 3Epidemiology and Prevention Unit-Department of Research, Fondazione IRCSS Istituto Nazionale dei Tumori di Milano, 20133 Milano, Italy; adalberto.cavaller@istitutotumori.mi.it (A.C.); elisabetta.venturelli@istitutotumori.mi.it (E.V.); arabella.mazzocchi@istitutotumori.mi.it (A.M.); eleonora.bruno@istitutotumori.mi.it (E.B.); franco.berrino@istitutotumori.mi.it (F.B.); patrizia.pasanisi@istitutotumori.mi.it (P.P.); 4Immunohematology and Transfusion Medicine Service-Department of Research, Fondazione IRCSS Istituto Nazionale dei Tumori di Milano, 20133 Milano, Italy; francesca.taverna@istitutotumori.mi.it; 5Chair Cancer Experimental Therapeutics, Department of Oncology Faculty of Health Science, McMaster University, Hamilton, ON L8V 1C3, Canada; muti@mcmaster.ca

**Keywords:** metformin, vitamin B_12_, holotranscobalamin II, methylmalonic acid

## Abstract

Metformin (MET) is currently being used in several trials for cancer prevention or treatment in non-diabetics. However, long-term MET use in diabetics is associated with lower serum levels of total vitamin B_12_. In a pilot randomized controlled trial of the Mediterranean diet (MedDiet) and MET, whose participants were characterized by different components of metabolic syndrome, we tested the effect of MET on serum levels of B_12_, holo transcobalamin II (holo-TC-II), and methylmalonic acid (MMA). The study was conducted on 165 women receiving MET or placebo for three years. Results of the study indicate a significant overall reduction in both serum total B_12_ and holo-TC-II levels according with MET-treatment. In particular, in the MET group 26 of 81 patients and 10 of the 84 placebo-treated subjects had B_12_ below the normal threshold (<221 pmol/L) at the end of the study. Considering jointly all B_12_, Holo-TC-II, and MMA, 13 of the 165 subjects (10 MET and 3 placebo-treated) had at least two deficits in the biochemical parameters at the end of the study, without reporting clinical signs. Although our results do not affect whether women remain in the trial, B_12_ monitoring for MET-treated individuals should be implemented.

## 1. Introduction

Metformin (MET) is the first-line treatment for type-2 diabetes and has been used for decades to treat this chronic condition [1]. Given its favorable effects on glycemic control, weight patterns, insulin requirements, and cardiovascular outcomes, MET has been recently proposed in addition to lifestyle interventions to reduce metabolic syndrome (MS) and age-related chronic diseases [2]. Observational studies have also suggested that diabetic patients treated with MET had a significantly lower risk of developing cancer or lower cancer mortality than those untreated or treated with other drugs [3,4]. For this reason, a number of clinical trials are in progress in different solid cancers.

One of the limitations in implementing long-term use of MET to prevent chronic conditions in healthy subjects relates to its potential lowering effect on vitamin B_12_ (B_12_). The aim of the present study was to assess the effect of three years of MET treatment in a randomized, controlled trial considering both B_12_ levels and biomarkers of its metabolism and biological effectiveness.

Cobalamin, also known as B_12_, is a water-soluble, cobalt-containing vitamin. All forms of B_12_ are converted intracellularly into adenosyl-Cbl and methylcobalamin—the biologically active forms at the cellular level [5]. Vitamin B_12_ is a vital cofactor of two enzymes: methionine synthase and L-methyl-malonyl-coenzyme. A mutase in intracellular enzymatic reactions related to DNA synthesis, as well as in amino and fatty acid metabolism. Vitamin B_12_, under the catalysis of the enzyme l-methyl-malonyl-CoA mutase, synthesizes succinyl-CoA from methylmalonyl-CoA in the mitochondria. Deficiency of B_12_, thus results in elevated methylmalonic acid (MMA) levels.

Dietary B_12_ is normally bound to proteins. Food-bound B_12_ is released in the stomach under the effect of gastric acid and pepsin. The free vitamin is then bound to an R-binder, a glycoprotein in gastric fluid and saliva that protects B_12_ from the highly acidic stomach environment. Pancreatic proteases degrade R-binder in the duodenum and liberate B_12_; finally, the free vitamin is then bound by the intrinsic factor (IF)—a glycosylated protein secreted by gastric parietal cells—forming an IF-B_12_ complex [6]. The IF resists proteolysis and serves as a carrier for B_12_ to the terminal ileum where the IF-B_12_ complex undergoes receptor (cubilin)-mediated endocytosis [7]. The vitamin then appears in circulation bound to holo-transcobalamin-I (holo-TC-I), holo-transcobalamin-II (holo-TC-II), and holo-transcobalamin-III (holo-TC-III). It is estimated that 20–30% of the total circulating B_12_ is bound to holo-TC-II and only this form is available to the cells [7]. Holo-TC-I binds 70–80% of circulating B_12_, preventing the loss of the free unneeded portion [6]. Vitamin B_12_ is stored mainly in the liver and kidneys.

Many mechanisms have been proposed to explain how MET interferes with the absorption of B_12_: diminished absorption due to changes in bacterial flora, interference with intestinal absorption of the IF–B_12_ complex (and)/or alterations in IF levels. The most widely accepted current mechanism suggests that MET antagonizes the calcium cation and interferes with the calcium-dependent IF–B_12_ complex binding to the ileal cubilin receptor [8,9]. The recognition and treatment of B_12_ deficiency is important because it is a cause of bone marrow failure, macrocytic anemia, and irreversible neuropathy [10].

In general, previous studies on diabetics have observed a reduction in serum levels of B_12_ after both short- and long-term MET treatment [1]. A recent review on observational studies showed significantly lower levels of B_12_ and an increased risk of borderline or frank B_12_ deficiency in patients on MET than not on MET [1]. The meta-analysis of four trials (only one double-blind) found a significant overall mean B_12_ reducing effect of MET after six weeks to three months of use [1]. A secondary analysis (13 years after randomization) of the Diabetes Prevention Program Outcomes Study, which randomized over 3000 persons at high risk for type 2 diabetes to MET or placebo, showed a 13% increase in the risk of B_12_ deficiency per year of total MET use [3]. In this study, B_12_ levels were measured from samples obtained in years 1 and 9. Stored serum samples from other time points, including baseline, were not available, and potentially informative red blood cell indices that might have demonstrated the macrocytic anemia, typical of B_12_ deficiency, were not recorded [3]. The HOME (Hyperinsulinaemia: the Outcome of its Metabolic Effects) study, a large randomized controlled trial investigating the long-term effects of MET versus placebo in patients with type 2 diabetes treated with insulin, showed that the addition of MET improved glycemic control, reduced insulin requirements, prevented weight gain but lowered serum B_12_ over time, and raised serum homocysteine, suggesting tissue B_12_ deficiency [4]. A recent analysis of 277 diabetics from the same trial showed that serum levels of MMA, the specific biomarker for tissue B_12_ deficiency [5], were significantly higher in people treated with MET than those receiving placebo after four years (on average) [4].

The risk of MET-associated B_12_ deficiency may be higher in older individuals and those with poor dietary habits. Prospective studies have found negative associations between obesity and B_12_ in numerous ethnicities [11,12]. An energy-dense but micronutrient-insufficient diet consumed by individuals who are overweight or obese might explain this [12]. Furthermore, obesity is associated with low-grade inflammation and these physiological changes have been shown to be associated, in several studies, with elevated C-reactive protein and homocysteine and with low concentrations of B_12_ and other vitamins [13,14].

As part of a pilot randomized controlled trial of the Mediterranean diet (MedDiet) and MET for primary prevention of breast cancer and other chronic age-related diseases in healthy women with tracts of MS [15] we tested the effect of MET on serum levels of B_12_, holo-TC-II, and MMA.

## 2. Material and Methods

### 2.1. Study Design

A pilot phase III randomized controlled trial of MedDiet and MET is ongoing at the Fondazione IRCCS Istituto Nazionale dei Tumori in Milan (Ethic approval code: INT76/08) for the primary prevention of breast cancer and other chronic age-related diseases in healthy peri-postmenopausal women with a waist circumference ≥85 cm and at least one other component of the metabolic syndrome reported below
-high plasma glucose (≥100 mg/100 mL)-high triglycerides (≥150 mg/100 mL)-low HDL cholesterol (<50 mg/100 mL)-systolic blood pressure ≥130 mm Hg or diastolic blood pressure ≥85 mm Hg.

The study design is a double-blind 2 × 2 factorial with 400 volunteers randomized in four groups of 100 each and allocated to the following treatments:-MET (1700 mg/day) + MedDiet intervention-Placebo + MedDiet intervention-MET (1700 mg/day) alone-Placebo alone.

At baseline, participants were asked to sign an informed consent form and attend an anthropometric visit, giving blood and urine samples. The baseline measurements served to check the tracts of the metabolic syndrome and the absence of exclusion criteria. The exclusion criteria were: diabetes, fasting glycemia more than 126 mg/dL in two repeated blood samples or concomitant treatment with MET, diagnosis of cancer in the last five years (except skin carcinomas), serum creatinine more than 124 micromol/L, proteinuria, concomitant treatment with potassium-sparing diuretics or proton pump inhibitors, and excessive alcohol consumption. The study lasted five years, with a MET/placebo treatment of three years on average. Once a year for the five years of the study each participant repeated the anthropometric visit and blood sampling, on approximately the same date as the previous one, as required by the study protocol.

The trial is now being extended to include 1600 women and men with MS (the Me.Me.Me. trial; ethic approval code: INT76/08) [16].

### 2.2. Participants

Among the 400 volunteers included in the trial, we selected 165 healthy women, aged 46–73 years, to test the effect of MET on serum levels of B_12_, holo-TC-II, and MMA. The sampled 165 women were selected on the basis of the following criteria: three years of MET or placebo treatment at the time of the analysis and availability of before and after treatment blood samples. Forty-four women were allocated to MET + MedDiet intervention, 37 to Placebo + MedDiet intervention, 37 received MET alone, and 47 Placebo alone.

### 2.3. Measurements

Blood samples were drawn at baseline and at each scheduled time points and sera were frozen at −70 °C and never thawed.

The B_12_ was determined on the baseline and after three years of samples using the Electro-Chemiluminescence Immuno Assay (ECLIA) method (COBAS E 601 autoanalyzer, Roche Diagnostic). The B_12_ measuring range was 50.0–2000 pg/mL. For internal quality control we used PreciControl Varia (Roche Diagnostic), two levels (normal and pathological). Furthermore, for the B_12_ tests, our laboratory participates in external quality certification programs organized by the Lombardy Region (Italy).

Quantitative determination of holo-TC-II (active-B_12_) was done using chemiluminescent microparticle immunoassay technology (Abbott Diagnostics, Wiesbaden, Germany). The measuring interval range of the Holo-TC-II assay (active-B_12_) was 5.0–128 pmol/L. Samples with values above 128 pmol/L were diluted with “architect multi-assay manual diluents” according to the manufacturer’s instructions. The performance of the instrument was evaluated using two levels, low and high, internal quality controls.

The MMA was determined by a liquid chromatographic separation and isotopic dilution mass spectrometric technique (ID-MSMS). The MMA measuring range was 6.25–200 ng/mL. A pool of plasma was used as internal quality control.

Inter-assay precision was investigated according to the Clinical and Laboratory Standards Institute (CLSI) protocol. The coefficients of inter-assay variation were respectively: <5.2% for total B_12_, <8% for holo-TC-II, and <7.9% for MMA.

### 2.4. Criteria and Source of Cut-Offs Used

#### 2.4.1. Vitamin B_12_ Assay

According to the World Health Organization (WHO), B_12_ status in adults is defined by the serum levels of the micronutrient with the following cut-off and definitions: ≥221 pmol/L is vitamin “B_12_ adequacy” (i.e., normal); between 148 and 221 pmol/L is “low B_12_”, and lower than 148 pmol/L is “B_12_ deficiency” (i.e., very low) [17,18]. Vitamin B_12_ levels <148 pmol/L have sensitivity exceeding 95% in patients with megaloblastic anemia [19,20], and this cut-off is most commonly used in research and clinical fields [20].

#### 2.4.2. Holo-TC-II Assay

The diagnostic accuracy of holo-TC-II remains controversial. The test is thought to have sensitivities and specificities comparable to that of serum B_12_ with regard to MMA elevations [5]. Several studies suggest that holo-TC-II slightly outperforms the B_12_ test [21,22]. The specificity of the holo-TC-II test remains unclear. According to the WHO guidelines, holo-TC-II levels are defined “normal” if higher than 50 pmol/L, “low” from 35 to 50 pmol/L and “very low” if lower than 35 pmol/L.

#### 2.4.3. MMA Assay

Vitamin B_12_, under the catalysis of the enzyme methylmalonyl-CoA mutase, synthesizes succinyl-CoA from methylmalonyl-CoA in the mitochondria. Deficiency of B_12_ thus results in elevated MMA levels. A high MMA test result has >95% sensitivity for B_12_ deficiency in patients with pernicious anemia [4]. Methylmalonic acid test cut-offs between 210 and 480 nmol/L are used to define B_12_ deficiency. The cut-off of ≥271 nmol/L is currently the most commonly used [23,24].

### 2.5. Statistical Analysis

After checking for the completeness and consistency of the data we employed two-way ANOVA to investigate the effect of treatment (placebo or MET) and/or that of diet (free or MedDiet) on the pivotal variables of interest (holo-TC-II, Vitamin B_12_ and MMA levels) on their original continuous scale. Specifically, in these models the response variable, i.e., the difference (Δ_T1–T0_) between the biomarker levels at baseline (T0) and after three years (T1), was modeled as a function of the main effect ‘‘treatment’’ (placebo or MET) and ‘‘diet’’ (free or MedDiet) and their first-order interaction term. The Fisher exact test was finally implemented to assess possible associations between the WHO’ deficit conditions and the four experimental groups.

All statistical analyses were performed with SAS software (Version 9.4.; SAS Institute, Inc., Cary, NC, USA), adopting a nominal significance level of α = 0.05.

## 3. Results

The subjects’ baseline anthropometric and clinical chemical parameters are shown in Appendix A, and hematologic variables at baseline and after three years, in Appendix A. There appeared to be no alterations to kidney and/or hematological function. Descriptive statistics for B_12_, holo-TC- II, and MMA at baseline (T0) and after three years (T1) are reported in Table 1.

The ANOVA results for both the total serum B_12_ levels and holo-TC-II absolute differences concentrations indicated significant decreases in the main effects for the treatment factor (*p* = 0.02 and *p* = 0.01, respectively). The diet factor, neither alone (*p* = 0.58 and *p* = 0.09) nor with treatment (*p* = 0.89 and *p* = 0.28), significantly contributed to the reductions in biomarkers.

Serum MMA showed no significant changes either in the main effects (*p* = 0.17 for treatment and *p* = 0.92 for diet) or in the interaction terms (*p* = 0.30). Figure 1 shows the interaction plots: with the levels of the treatment factor (placebo and MET) on the *x*-axis and the mean values of the response variables (a) B_12_, (b) holo-TC-II, and (c) MMA on the *y*-axis, with separate lines (continuous and dotted) for the two levels of the diet factor (Free and MedDiet).

On the basis of WHO guidelines, we examined the sample’s deficit conditions at the end of the interventions (Table 2): WHO guidelines on B_12_ status define “adequate” B_12_ values as higher or equal to 221 pmol/L (i.e., normal), “low” from 148 to 221 and “B_12_ deficiency” if lower than 148 (i.e., very low). Holo-TC-II levels are defined “normal” if higher than 50 pmol/L, “low” from 35 to 50, and “very low” if lower than 35. Methylmalonic acid levels are considered “normal” if below the cutoff of 271 nmol/L. As regards B_12_, 26 of 81 MET-treated patients and 10 of 84 placebo-treated subjects had levels below the normal threshold meaning that around 22% of the total study population did not have adequate total B_12_ levels at the end of the study. Holo-TC-II levels were “low” or “very low” in 16 subjects, 11 MET-treated and five placebo-treated. Methylmalonic acid levels were above the normal threshold in less than 7% of the MET treated patients and in one placebo-treated subject. As reported in Table 2, no associations were observed between WHO’s deficit conditions and the four experimental groups for both Holo-TC-II and MMA levels (Fisher exact test, *p*-value: 0.63, 0.40, respectively). On the contrary, for B_12_, a statistically significant association was observed Fisher exact test, *p*-value: 0.02).

Considering jointly all the biochemical parameters (B_12_, holo-TC-II and MMA), 13 subjects (10 MET- and 3-placebo-treated) out of 165 had at least two deficits in the biochemical parameters at T1. However, patients reported no clinical signs.

## 4. Discussion

In line with previous studies in diabetics, our data indicate that long-term use of MET lowers circulating B_12_ levels in healthy women with tracts of MS. The decline was significant in all groups of MET-treated subjects regardless of the MedDiet intervention. Almost a third (32%) of the women in the trial presented B_12_ deficiency (<221 pmol/L) after three years of MET daily doses of 1700 mg: these results were similar to those from other studies (8.6%–31%) [1,2,3].

Metformin is the most widely prescribed oral medication for type-2 diabetes. Given its potential anti-cancer effect [25,26,27,28,29,30,31], several trials are ongoing in non-diabetics now too [15,16,32]. Our results are a novelty because they are the first ones including women free of acute and/or chronic diseases. The aim of the present study was to check safety, evaluating the effect of MET in term of its effects over time on serum levels of B_12_, holo-TC-II, and MMA in women under treatment for three years. For the same purpose, we also checked the participants’ blood counts for any sign of macrocytic anemia, as well as their hemoglobin and hematocrit. We considered it essential to confirm that long-term treatment with 1700 mg/day of MET, a potential “preventive” drug, is completely safe.

Despite the reduction in B_12_ levels, our results do not suggest the on-set of clinical B_12_ deficiency nor any clinical effect that might influence whether women remain in the trial. A recent meta-analysis about B_12_ deficiency (comprising 31 studies) showed that, compared to diabetics not taking MET, patients taking the drug had a significantly higher risk of B_12_ deficiency and lower serum B_12_ concentration, in a duration- and dose-dependent manner, without any significant association between MET use and prevalence of anemia [33]. Furthermore, subgroup analyses indicated that only patients with a mean of more than three years of MET therapy or patients on mean daily doses of >2000 mg had a significantly higher risk of B_12_ deficiency than patients not taking MET. In this larger meta-analysis, the deficiency was diagnosed using only the direct measurement of circulating B_12_ which, however, is not considered to sufficiently reflect its metabolic status [34].

The strength of our study is the “global” evaluation of the B_12_ status of participants, also taking into account serum levels of holo-TC-II and MMA. Holo-TC-II was below the cut-off (≤50 pmol/L) in 11 women (14%) after the third year of MET treatment. The MMA levels were above the normal threshold in less than 7% of the MET-treated patients and one placebo-treated subject.

Herrmann et al. [35] reported that MMA assay leads to a three-block classification of patients, where: (1) MMA < 271 with holo-TC-II < 35 represents a negative B_12_ balance (insufficiency); (2) MMA < 271 with holo-TC-II 36–50 suggests the patient is unlikely to be B_12_-deficient; and (3) a range of possible B_12_-deficient patients have MMA ≥ 271 and low or very low holo-TC-II. According to this classification, it is evident that only holo-TC-II gives important information about B_12_ status because MMA rises only after the declines in B_12_ and holo-TC-II become important (very low). In our series, five MET-treated subjects had high MMA ≥ 271 but only one had low B_12_ and holo-TC-II. Furthermore, in the placebo group, just one subject had an increase only in MMA.

Considering all the biochemical parameters together (B_12_, holo-TC-II, and MMA), only 13 subjects (10 MET- and 3-placebo-treated) out of 165 had at least two deficiencies in T1. However, no clinical signs were found in hematologic data or reported by patients. This small number might be attributable to the large amount of B_12_ in the human body—approximately 2000–4000 μg (stored primarily in the liver). Roughly 1 μg of B_12_ is consumed every day; thus, many years (at least ten) are necessary to reach a deficit status, which helps explain why we detected so few cases of biochemical deficiency after three years. It is reasonable to believe that even longer is needed to detect a clinical B_12_ deficiency. Furthermore, the cause of MET-induced B_12_ deficiency is not known, but it appears to involve altered uptake of the cubilin receptor, which reduces B_12_ absorption in the ileum.

Another mechanism has been hypothesized for the low serum B_12_ levels during MET use. Several groups have noted that in rodent models, MET treatment increases the accumulation of B_12_ in the liver, lowering circulating vitamin B_12_ levels, suggesting that MET alters the tissue distribution and metabolism of B_12_, rather than causing true deficiency. This novel mechanism might also explain the absence of clinical signs despite the apparently very low levels of B_12_; thus, the definition of vitamin deficiency, based only on serum levels, might be incorrect. Holo-TC-II assay might give more valuable information, although true vitamin deficiency can be definitively diagnosed only on the basis of an increase in MMA and the presence of clinical conditions (megaloblastic anemia, neuropathies).

Finally, some notes about the possible limitations of the current study should be mentioned. The current study considered all the women treated with MET or placebo for three years at the time of the analysis and with both baseline and 3-year blood samples available. Further evaluations considering also additional time points could provide additional information about MET long-term treatment on B_12_ reduction.

Another limit is that we were not able to study dietary components. Our results did not suggest any contribution of the MedDiet intervention to the B_12_ reduction, either alone or in combination with MET. Unfortunately, we did not use repeated dietary assessment methods to check compliance with the diet. In fact, according to the trial design, the main method for assessing dietary differences in the MedDiet and FreeDiet group was based on anthropometry and clinical chemistry, and 24 h food frequency diaries concerning food eaten the previous day were scheduled at baseline and at the end of the five-year intervention.

Current guidelines do recognize the risk of B_12_ deficiency as a disadvantage of MET treatment for type 2 diabetes [21]. However, this is rarely investigated in diabetics. We found that after three years of treatment the frequency of MET-induced B_12_ biochemical deficiency was low and the cases detected had no clinical-hematological signs. Given these results, three years of MET treatment (as provided by our trial protocol) do not seem to be associated with clinical deficiency of B_12_. However, given the growing population of individuals who receive MET for other indications (including high risk of diabetes, polycystic ovary syndrome, metabolic syndrome, cancer prevention), and the chronic nature of these treatments, we feel that monitoring of B_12_ (and Holo-TC-II, if B_12_ levels are low-normal) in MET-treated individuals should be considered routinely before starting and during treatment.

## Figures and Tables

**Figure 1 nutrients-11-01020-f001:**
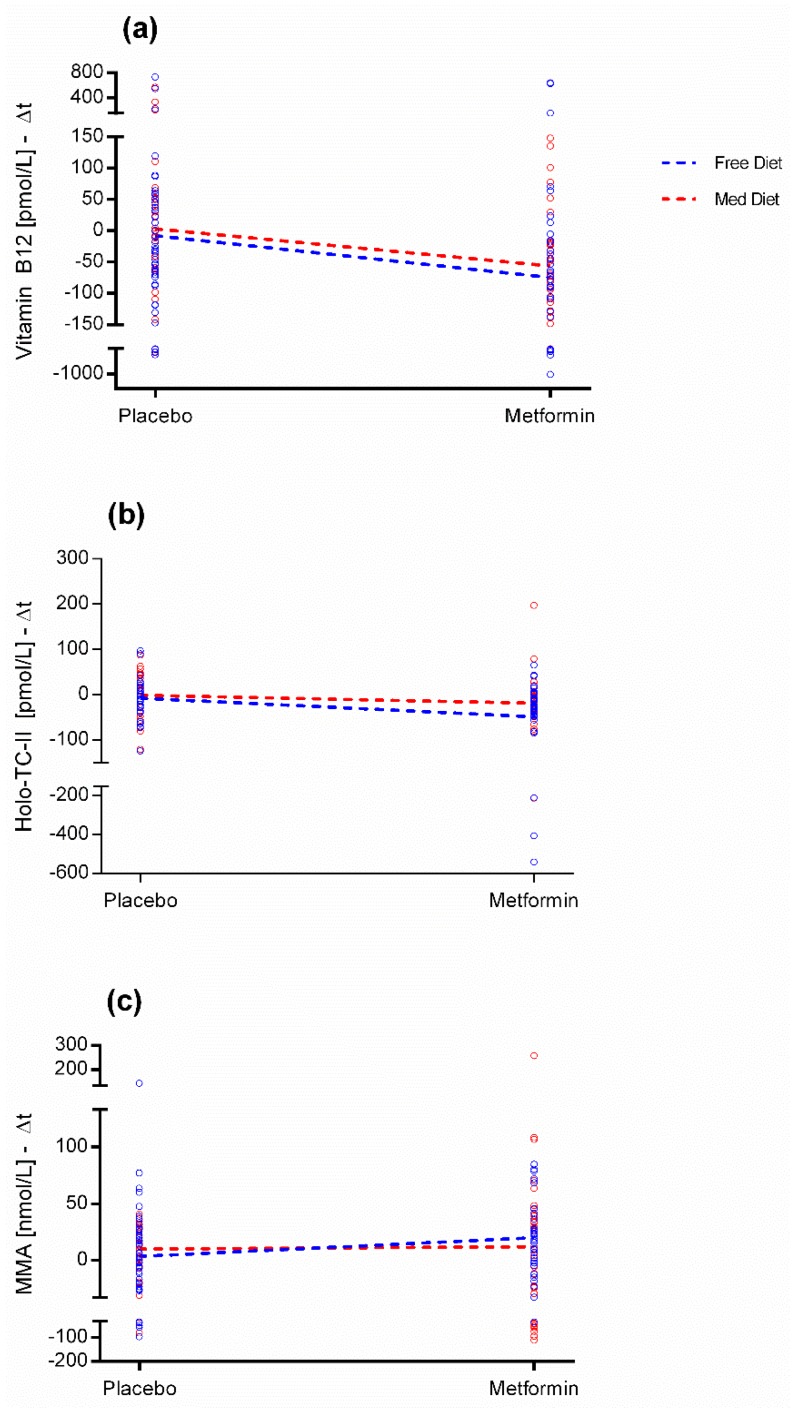
Interaction plots. Each graph depict the ANOVA results on (**a**) B_12_, (**b**) Holo-transcobalamin-II (Holo-TC-II), and (**c**) Methylmalonic acid (MMA). They show the average response variable (i.e., absolute differences between baseline and after three years on the *y*-axis, Δt = T1–T0) in relation to the treatment level (on the *x*-axis) for each level of diet. Each dot represents the individual difference between baseline and the three years levels of each considered variable.

**Table 1 nutrients-11-01020-t001:** Descriptive statistics of the variables investigated.

	Group	Time	*n*	Min	25th Centile	Median	75th Centile	Max	IQR
B_12_ [pmol/L]	M-MedDiet	baseline	44	143.7	276.4	335.3	402.0	756.9	125.7
		3rd year	44	120.9	205.3	262.4	354.9	765.7	149.6
	M-FreeDiet	baseline	37	150.3	268.3	350.1	470.9	1474.0	202.7
		3rd year	37	109.8	193.1	284.5	420.8	1474.0	227.7
	P-MedDiet	baseline	37	136.3	260.2	316.9	353.0	817.3	92.9
		3rd year	37	147.4	244.7	299.2	381.0	844.6	136.3
	P-FreeDiet	baseline	47	195.3	287.4	358.2	426.7	642.7	139.3
		3rd year	47	46.4	255.7	330.9	412.7	1312.6	157.0
Holo-TC-II [pmol/L]	M-MedDiet	baseline	44	34.0	85.5	112.5	140.0	402.0	54.5
		3rd year	44	22.1	61.5	84.0	117.5	416.0	56.0
	M-FreeDiet	baseline	37	47.0	91.0	109.0	152.0	780.0	61.0
		3rd year	37	34.0	66.0	83.0	119.0	246.0	53.0
	P-MedDiet	baseline	37	58.0	89.0	100.0	121.0	248.0	32.0
		3rd year	37	48.0	85.0	108.0	126.0	192.0	41.0
	P-FreeDiet	baseline	47	49.0	88.0	115.0	138.0	248.0	50.0
		3rd year	47	41.0	80.0	112.0	140.0	238.0	60.0
MMA [nmol/L]	M-MedDiet	baseline	44	83.9	105.5	132.6	182.6	359.3	77.1
		3rd year	44	78.0	120.8	139.4	195.7	411.8	75.0
	M-FreeDiet	baseline	37	71.2	106.8	125.4	156.8	341.5	50.0
		3rd year	37	89.8	119.5	144.9	170.3	422.0	50.8
	P-MedDiet	baseline	37	69.5	92.4	106.8	145.8	221.2	53.4
		3rd year	37	75.4	99.1	126.3	150.0	244.1	50.8
	P-FreeDiet	baseline	47	83.9	105.9	131.3	163.5	258.5	57.6
		3rd year	47	72.0	105.9	137.3	163.5	377.9	57.6

Abbreviations: *n* = number, IQR = Inter Quartile Range, Holo-TC-II = Holo-transcobalamin-II, MMA = Methylmalonic acid.

**Table 2 nutrients-11-01020-t002:** Frequency distribution of B_12_ deficit status at T1 according to World Health Organization WHO guidelines, for each treatment group.

	Total		B_12_						Holo-TC-II						MMA			
			Normal		Low		Very Low		Normal		Low		Very Low		Normal		Pathological	
	*n*	%	*n*	%	*n*	%	*n*	%	*n*	%	*n*	%	*n*	%	*n*	%	*n*	%
M-MedDiet	44	54.3	30	37.0	12	14.8	2	2.5	37	45.7	5	6.2	2	2.5	41	50.6	3	3.7
M-FreeDiet	37	45.7	25	30.9	9	11.1	3	3.7	33	40.7	3	3.7	1	1.2	35	43.2	2	2.5
total	81	100.0	55	67.9	21	25.9	5	6.2	70	86.4	8	9.9	3	3.7	76	93.8	5	6.2
P-MedDiet	37	44.0	32	38.1	4	4.8	1	1.2	35	41.7	2	2.4	0	0.0	37	44.1	0	0.0
P-FreeDiet	47	56.0	42	50.0	2	2.4	3	3.6	44	52.4	3	3.6	0	0.0	46	54.8	1	1.2
total	84	100.0	74	88.1	6	7.1	4	4.8	79	94.0	5	6.0	0	0.0	83	98.8	1	1.2

Abbreviations: M = Metformin, P = Placebo, MedDiet = Mediterranean diet, Holo-TC-II = Holo-transcobalamin-II, MMA = Methylmalonic acid. Percentages were computed with respect to the total number of subjects treated with metformin or placebo, respectively.

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
