# Peer review of "Monitoring Vitamin B12 in Women Treated with Metformin for Primary Prevention of Breast Cancer and Age-Related Chronic Diseases"

_nutrients, 2019, doi:10.3390/nu11051020_

Round 1
Reviewer 1 Report
This manuscript describes the status of vitamin B12 in a pilot study of women with indicators of metabolic syndrome being followed longitudinally for breast cancer prevention by a Mediterranean diet and/or metformin treatment. The point of the B12 analysis is to ascertain the potential risk of a metformin-induced decrease in B12 for the study cohort. The authors found evidence of a decrease in B12 status but without apparent clinical manifestations.
Being a pilot study, the number of subjects is quite limited. Nonetheless, ANOVA analysis revealed an adverse influence of metformin on plasma B12 and Holo TCII levels, but not MMA levels at the 3-year timepoint. The absence of statistical analysis for Table 2 data is puzzling and should be addressed. While the results clearly indicate a metformin-induced decreased in B12 status, associated clinical symptoms were not observed at the 3-year timepoint. However, the statement on line 210: "....patients reported no clinical signs." suggests that there was no active attempt to examine the occurrence of clinical signs (e.g. hematological measurements). Moreover, homocysteine levels were not measured. The concluding statement: "We are confident that our intervention strategy for primary prevention of chonic diseases and conditions is not associated with clinical deficiency of B12 for at least three year of MET treatment" is accordingly not justified. It seems clear that this is the conclusion the authors wished for in order to justify their ongoing clinical study.
Author Response
Reviewer 1
This manuscript describes the status of vitamin B12 in a pilot study of women with indicators of metabolic syndrome being followed longitudinally for breast cancer prevention by a Mediterranean diet and/or metformin treatment. The point of the B12 analysis is to ascertain the potential risk of a metformin-induced decrease in B12 for the study cohort. The authors found evidence of a decrease in B12 status but without apparent clinical manifestations.
Being a pilot study, the number of subjects is quite limited. Nonetheless, ANOVA analysis revealed an adverse influence of metformin on plasma B12 and Holo TCII levels, but not MMA levels at the 3-year timepoint.
The absence of statistical analysis for Table 2 data is puzzling and should be addressed. While the results clearly indicate a metformin-induced decreased in B12 status, associated clinical symptoms were not observed at the 3-year timepoint.
R: we thank the Reviewer for his/her comment. Please consider that our principal aim was to evaluate changes during time in vitamin B12, Holo-TC-II and MMA by working on their original continuous scale. Accordingly, due to the underline nature of the design (2x2 factorial) we implemented an ANOVA analysis to evaluate significant changes in the level of the pivotal variable under investigation. Moreover, for descriptive purpose, we evaluate the deficit status with regards to the WHO criteria. Following the Reviewer comment, we performed an association analysis to evaluate relationships between the deficit status at T1 and the four experimental groups. Results of this supplementary analysis are reported in the results section. Coherently we also updated the statistical analysis section.
However, the statement on line 210: "....patients reported no clinical signs." suggests that there was no active attempt to examine the occurrence of clinical signs (e.g. hematological measurements).
R: we thank the Reviewer for his/her comment. We have already produced two supplementary tables (1S and 1S2) with the biological informations requested by the reviewer but, unfortunately, these tables were probably not sent with the manuscript. We are now reporting below these supplementary tables.
Table S1. Baseline anthropometric parameters
n. | min | 25th centile | median | 75th centile | max | IQR | |
Age (yrs) | 165 | 46.6 | 53 | 57.8 | 63.3 | 73 | 10.3 |
Height (cm) | 165 | 144 | 155 | 159 | 164 | 173 | 9 |
Weight (kg) | 165 | 57.7 | 68 | 74.2 | 83.3 | 132.3 | 15.3 |
Waist circumference | 165 | 85 | 87 | 93 | 99.5 | 133 | 12.5 |
IQR=interquartile range (75th centile-25th centile)
Table S2. Hematologic variables
time | n. | min | 25th centile | median | 75th centile | max | IQR | |
Serum creatinine (mg/dL) | baseline | 165 | 0.5 | 0.7 | 0.8 | 0.8 | 1.2 | 0.1 |
3rd year | 163 | 0.5 | 0.7 | 0.7 | 0.8 | 1.2 | 0.2 | |
Hematocrit | baseline | 165 | 28.5 | 40.3 | 42.2 | 44.0 | 49.1 | 3.7 |
3rd year | 164 | 34.0 | 40.2 | 41.8 | 43.5 | 46.5 | 3.3 | |
Erythrocytes (x106) | baseline | 165 | 4.1 | 4.6 | 4.8 | 5.0 | 7.9 | 0.4 |
3rd year | 164 | 3.9 | 4.5 | 4.7 | 5.0 | 6.1 | 0.4 | |
Hemoglobin (g/dL) | baseline | 165 | 9.7 | 13.3 | 13.9 | 14.5 | 15.7 | 1.2 |
3rd year | 164 | 10.1 | 13.2 | 13.8 | 14.5 | 15.4 | 1.3 |
IQR=interquartile range (75th centile-25th centile)
Moreover, homocysteine levels were not measured.
R: we thank the Reviewer for his/her comment. It is well known that homocysteine (HC) levels are associated with vitamin B12 deficiency, but it is also known that high MMA test result has > 95% sensitivity for B12 deficiency in patients with pernicious anemia [1]. MMA is clearly more specific to vitamin B-12 deficiency compared to HC (2.) However, elevated MMA and HC levels together have been found to be 99.8% sensitive for diagnosing functional vitamin B-12 deficiency [3], even when serum vitamin B12 levels are within the reference values. Performing both tests would have been too expensive for our study, and so we opted for MMA the best test between of the two.
1. Out, M.; Top, W. M. C.; Lehert, P.; Schalkwijk, C. A.; Stehouwer, C. D. A.; Kooy, A. Long-term treatment with metformin in type 2 diabetes and vitamin D levels: A post-hoc analysis of a randomized placebo-controlled trial. Diabetes Obes. Metab 2018,20,1951-1956
2. Oh R, Brown DL. Vitamin B12 deficiency. Am Fam Physician 2003; 67(5):979–986
3. Savage DG, Lindenbaum J, Stabler SP, Allen RH. Sensitivity of serum methylmalonic acid and total homocysteine determinations for diagnosing cobalamin and folate deficiencies. Am J Med 1994; 96(3):239–246.
The concluding statement: "We are confident that our intervention strategy for primary prevention of chonic diseases and conditions is not associated with clinical deficiency of B12 for at least three year of MET treatment" is accordingly not justified. It seems clear that this is the conclusion the authors wished for in order to justify their ongoing clinical study.
R: we thank the Reviewer for his/her comment. According to our trial protocol (lines 125-126 of the study design section) the MET/placebo treatment last three years on average. As showed by the supplementary table 2 (S2 ) we did not observe clinic-hematological signs of B12 deficiency in our population after three years of MET treatment and we are confident that our treatment is safe. However, we have modified the concluding statement according to the reviewer’s comment.
Reviewer 2 Report
The article represents the clinical trial of Metformin on serum levels of B12, holo-transcobalamin II (holo-TC-II), and methylmalonic acid (MMA). The statistical analysis of the data is adequate.
Author Response
Reviewer 2
It is well known that vitamin B12 measurement has low diagnostic sensitivity for vitamin B12 deficiency and other markers such as MMA is more sensitive and specific for the diagnosis. However, in this study, the vitamin B12 measurement showed the highest sensitivity among the biochemical parameters. Given the limitation of vitamin B12 assays, possible methodological problem or other influencing factors should be investigated and explained.
R: we thank the Reviewer for this point. In our study, the diagnostic sensitivity of the B12 assay alone is not associated with a clinical deficit. Although there is still no consensus, many authors agree that vitamin B12 deficiency can only be defined by considering at least 2 biomarkers with altered values (B12 and Holo and MMA or homocysteine). Furthermore, no other known pre-analytical factor influenced our dosage; in fact the samples examined were excellent without hemolysis or lipemia problems (triglycerides<400mg/dl) and participants into the trial did not take mineral or vitamin supplements such as biotin, the main interferer in dosages performed with the ECLIA method.
Line 145-8
There is a lack of information on analytical performance of the assay methods. The authors mentioned only inter-assay precision. What about analytical measurement range, internal and external quality control?
R: we thank the Reviewer for his/her comment. We significantly improved the sub-section “2.3. Measurements” accordingly.
Line 149-168
Please delete inappropriate content for “Method” section and briefly describe the cut-offs used in this study.
R:we agree. We added a specific sub-section “2.4. Criteria and source of cut-offs used” according to the reviewer suggestion.
Table 2
It is not very informative to show all the descriptive statistics (min/max, 25/57/75 percentile, IQR). Please include only key statistics with P values.
R:Proabably there is a misleading. Table 2 reports the Frequency distribution of B12 deficit status at T1 according to WHO guidelines, for each experimental group. If Reviewer are referring to the descriptive Table 1, we prefer to leave all the main descriptive quantities (min/max, 25/50/75 percentile, IQR) to provide a comprehensive overview of the data distribution. According to the underline nature of the experimental design (2x2 factorial design) we implemented an ANOVA analysis to evaluate relationships between changes in the level of the pivotal variables (T1-T0 difference) and the Diet and Treatment factor: thus, only p-value arising from the ANOVA analysis are reported in the results section. Please consider that in reply to Reviewer 1 we performed an additional analysis starting from data reported in Table 2.
Fig. 2.
Please consider changing the presentation style using scatter plot rather than just showing the average values.
R: Probably there is a misleading as no Figure 2 are reported in the paper. If Reviewer is referring to Figure 1, the graphs (i.e. interaction plots) depict the ANOVA results by showing the average response variable in relation to the treatment level for each level of diet. To aid the reader in the interpretation of the results, we updated Figure 1, by displaying each individual observation (i.e. dots).
Reviewer 3 Report
It is well known that vitamin B12 measurement has low diagnostic sensitivity for vitamin B12 deficiency and other markers such as MMA is more sensitive and specific for the diagnosis. However, in this study, the vitamin B12 measurement showed the highest sensitivity among the biochemical parameters. Given the limitation of vitamin B12 assays, possible methodological problem or other influencing factors should be investigated and explained.
Line 145-8
There is a lack of information on analytical performance of the assay methods. The authors mentioned only inter-assay precision. What about analytical measurement range, internal and external quality control?
Line 149-168
Please delete inappropriate content for “Method” section and briefly describe the cut-offs used in this study.
Table 2
It is not very informative to show all the descriptive statistics (min/max, 25/57/75 percentile, IQR). Please include only key statistics with P values.
Fig. 2.
Please consider changing the presentation style using scatter plot rather than just showing the average values.
Author Response
Reviewer 3
It is well known that vitamin B12 measurement has low diagnostic sensitivity for vitamin B12 deficiency and other markers such as MMA is more sensitive and specific for the diagnosis. However, in this study, the vitamin B12 measurement showed the highest sensitivity among the biochemical parameters. Given the limitation of vitamin B12 assays, possible methodological problem or other influencing factors should be investigated and explained.
R: we thank the Reviewer for this point. In our study, the diagnostic sensitivity of the B12 assay alone is not associated with a clinical deficit. Although there is still no consensus, many authors agree that vitamin B12 deficiency can only be defined by considering at least 2 biomarkers with altered values (B12 and Holo and MMA or homocysteine). Furthermore, no other known pre-analytical factor influenced our dosage; in fact the samples examined were excellent without hemolysis or lipemia problems (triglycerides<400mg/dl) and participants into the trial did not take mineral or vitamin supplements such as biotin, the main interferer in dosages performed with the ECLIA method.
Line 145-8
There is a lack of information on analytical performance of the assay methods. The authors mentioned only inter-assay precision. What about analytical measurement range, internal and external quality control?
R: we thank the Reviewer for his/her comment. We significantly improved the sub-section “2.3. Measurements” accordingly.
Line 149-168
Please delete inappropriate content for “Method” section and briefly describe the cut-offs used in this study.
R:we agree. We added a specific sub-section “2.4. Criteria and source of cut-offs used” according to the reviewer suggestion.
Table 2
It is not very informative to show all the descriptive statistics (min/max, 25/57/75 percentile, IQR). Please include only key statistics with P values.
R:Proabably there is a misleading. Table 2 reports the Frequency distribution of B12 deficit status at T1 according to WHO guidelines, for each experimental group. If Reviewer are referring to the descriptive Table 1, we prefer to leave all the main descriptive quantities (min/max, 25/50/75 percentile, IQR) to provide a comprehensive overview of the data distribution. According to the underline nature of the experimental design (2x2 factorial design) we implemented an ANOVA analysis to evaluate relationships between changes in the level of the pivotal variables (T1-T0 difference) and the Diet and Treatment factor: thus, only p-value arising from the ANOVA analysis are reported in the results section. Please consider that in reply to Reviewer 1 we performed an additional analysis starting from data reported in Table 2.
Fig. 2.
Please consider changing the presentation style using scatter plot rather than just showing the average values.
R: Probably there is a misleading as no Figure 2 are reported in the paper. If Reviewer is referring to Figure 1, the graphs (i.e. interaction plots) depict the ANOVA results by showing the average response variable in relation to the treatment level for each level of diet. To aid the reader in the interpretation of the results, we updated Figure 1, by displaying each individual observation (i.e. dots).